# The prevalence of non-communicable diseases in northwest Ethiopia: survey of Dabat Health and Demographic Surveillance System

Solomon Mekonnen Abebe,[1] Gashaw Andargie,[2] Alemayehu Shimeka,[3] Kassahun Alemu,[3] Yigzaw Kebede,[3] Mamo Wubeshet,[4] Amare Tariku,[1] Abebaw Gebeyehu,[5] Mulugeta Bayisa,[3] Mezgebu Yitayal,[2] Tadesse Awoke,[3] Temesgen Azmeraw,[5] Melkamu Birku[5,6,7]

For numbered affiliations see end of article.

**Correspondence to**
Dr Solomon Mekonnen Abebe; solomekonnen@yahoo.com

## ABSTRACT

**Objective** The main objective of this study was to investigate the magnitude and associated factors of non-communicable chronic diseases (NCDs) at the Dabat Health and Demographic Surveillance System (DHDSS) site in the northwestern part of Ethiopia.

**Design** A population-based cross-sectional study was conducted from October to December 2014.

**Setting** HDSS site, Ethiopia.

**Participants** A total population of 67 397 living in 16 053 households was included in the study.

**Measures** Structured interviewer-administered questionnaire was used to collect data. Self-reported morbidity was used to ascertain NCD. A binary logistic regression model was employed to identify the determinants of NCDs.

**Result** One thousand one hundred sixty (1.7%) (95% CI 1.62 to 1.82) participants were found with at least one type of NCD. Heart disease and hypertension which accounted for 404 (32.2%) and 401 (31.9%), of the burden, respectively, were the most commonly reported NCDs, followed by 347 (27.7%) asthma, 62 (4.9%) diabetes mellitus and 40 (3.2%) cancer cases. Advanced age (≥65 year) (adjusted OR (AOR)=19.6; 95% CI 5.83 65.70), urban residence (AOR=2.20; 95% CI 1.83 to 2.65), household food insecurity (AOR=1.71; 95% CI 1.37 to 2.12) and high income (AOR=1.28; 95% CI 1.02 to 1.59) were significantly associated with the reported history of NCDs, whereas low (AOR=0.36; 95% CI 0.31 to 0.42) and moderate (AOR=0.33; 95% CI 0.22 to 0.48) alcohol consumption, farming occupation (AOR=0.72; 95% CI 0.57 to 0.91), and work-related physical activities (AOR=0.66; 95% CI 0.50 to 0.88) were inversely associated with NCDs.

**Conclusion** There is a high burden of NCDs at the Dabat HDSS site. Promotion of regular physical exercise and reducing alcohol consumption are essential to mitigate the burden of NCDs. In addition, preventive interventions of NCDs should be strengthened among urban dwellers, older age people and people of higher economic status.

## Strengths and limitations of this study

► This is the first community-based prevalence study examining the magnitude and associated factors of non-communicable chronic diseases (NCDs) at a health and demographic surveillance site in Ethiopia.

► It has stringently included all age groups.

► It deals not only with the prevalence of NCDs but also with factors associated with it. Our findings provide important information that can be used to improve screening and care through medical evaluation to detect and manage NCDs.

► One limitation of this study is the use of self-report of NCDs that may introduce recall and social desirability biases and the morbidities reported might not be accurate diagnosis of disease conditions.

► Another limitation is that the study used only a cross-sectional design which might not show temporal relationships; thus, the observed associations might not necessarily be causal.

## INTRODUCTION

The burden of non-communicable chronic diseases (NCDs) is rising rapidly and has now become a major challenge to global development.[1] Globally, it is the leading cause of death, killing more people than all other causes each year. Low-income and middle-income countries are also disproportionately suffering from the consequences of these diseases.[2]

There are significant social, economic and health losses as a result of the increasing prevalence of NCDs. Even though crises due to NCDs can be deterred through cost-effective and feasible interventions, around 80% of the deaths in these countries are caused by these diseases.[3]

Non-communicable diseases have not received equal attention with communicable diseases in middle-income and low-income countries.[4] There is inadequate information with regard to the epidemiology of NCDs in developing countries for governments to

BMJ

develop NCD-related action plans, leaving major policy gaps for intervention. Hence, country-specific epidemiological and experimental studies are required for a better understanding of the burden and causes of such particular NCDs, as diabetes mellitus (DM), cardiovascular diseases (CVD), hypertension (HTN), cancers and chronic respiratory diseases .[2]

In Ethiopia, where health outcomes are highly dependent on cultural values, strong research is needed to properly assess lifestyle and anthropological measures to reveal the causes, prevention and control of NCDs. The country is in a major economic transition, bringing new ways of life to its population. As a result, there is also an epidemiological transition in terms of diseases from infectious to NCDs attributable to modified risk factors.[5] Some of the modifiable risk factors are physical inactivity, unhealthy diet, increasing alcohol and tobacco consumption. To halt the increasing prevalence of these diseases, a comprehensive multidisciplinary research is vital to elucidate the root causes and trends in the epidemiological transition

Moreover, the Ethiopian healthcare system is not designed for chronic healthcare, whose costs are huge. NCDs add to the nation's burden of poverty, slow down development and increase health inequities, which are believed to impose a huge demand on the healthcare system, creating economic pressure in the country. In Ethiopia, NCD prevention and control remain inadequate despite the growing burden of the problem. The few studies conducted in Ethiopia are inconclusive in revealing the epidemiology of the diseases.[6] Besides, almost all of the studies done are institution-based and do not expose the prevalence at the population level. This has contributed to policy gaps, leading to inadequate government investment on prevention and treatment of NCDs.[7]

To understand the true burden of NCDs in Ethiopia, population-based surveys are essential. Besides, it is time to assess the magnitude of NCDs and its associated factors at the population level for easy and cost-effective prevention.

## OBJECTIVE
The main objective of this study was to assess the magnitude and associated factors of NCDs at a Dabat Health and Demographic Surveillance System (DHDSS) site in the northwestern part of the country.

## METHODS
### Study Design
A population-based cross-sectional study was conducted from October to December 2014 at the Dabat Health and Demographic Surveillance System (HDSS) site, Dabat District, northwest Ethiopia.

### Setting
The district is located 820 km from Addis Ababa, the capital of Ethiopia. Based on the 2007 national census conducted by the Central Statistical Agency (CSA),[8] the district has a total projected population of 216 240 in 1187.93 sq. km of land. Dabat has a population density of 122.49, which is greater than the zonal average of 63.76 persons per sq. km.

The Dabat HDSS site was launched in the rural district of Dabat, by the Gondar College of Medical and Health Sciences (now University of Gondar) in 1995. In Ethiopia, there is inadequate vital events' registration system to date; instead, population and housing census, national sample surveys and health facility reports are the main sources of information on population health. Thus, Dabat HDSS was established in order to produce a longitudinal data to aid in better health policy formulation, informed decision-making and health practice.

### Participants
At the moment, 67 397 people live in the 16 053 households of the randomly selected 13 (9 rural and 4 urban) kebeles (the lowest administrative units) of the DHDSS site. Household level data on the general sociodemography of family members, housing and environmental conditions, food security, chronic diseases, risk factors for non-communicable diseases, injury in the last 2 weeks, maternal and child health are updated every 6 months, that is, twice a year. All permanent residents in the DHDSS area were eligible to participate in the study. The majority of the residents live on subsistence agriculture, producing mainly food crops.

### Data sources/measurement
A structured, interviewer administered questionnaire was used to collect data on reported history of blood pressure, diabetes, cancer, heart disease, asthma and history of other risk factors for chronic disease conditions (alcohol consumption, cigarette smoking, vegetable consumption and physical activity). Data were collected by diploma graduate professionals who were experienced in data collection at the DHDSS Research Center. The questionnaire was developed by considering questions previously used at this site. The English version of the questionnaire was translated to Amharic and back translated to English by language experts to maintain the consistency of meaning.

### Sample size and sampling technique
All permanent residents (67 397) of the HDSS site living in 16 053 households were included in the study. This study included all the permanent residents at Dabat DHDSS Research Center. The heads of households, mostly mothers, were interviewed to capture data on events (NCDs) that happened in the families. The outcome of this study, NCD, was ascertained using reported history of having at least one of the NCDs, such as DM, HTN, cancer, asthma and heart disease prior to data collection. Accordingly, a household member was labelled as having non-common chronic diseases if they had at least one reported NCD type. Sociodemographic characteristics

(age, sex, ethnicity, religion, marital status, residence, wealth index and family size), food security status and other health-related factors were considered as independent variables.

## Quantitative variables

Food security is defined as existing when 'all people, at all times, have physical and economic access to sufficient, safe and nutritious food to meet their dietary needs and food preferences for an active and healthy life.' In order to assess the food security status of the households, an 18-item community food insecurity access scale assessment tool was adopted from the Household Food Insecurity Access Scale (HFIAS) Indicator Guide V.3 and categorised into four levels, using the HFIS variables. A frequency of occurrence question was asked to determine whether the condition happened rarely (1 or 2 times), sometimes (3 to 10 times) or often (>10 times) in the past 4 weeks, to find out whether participants were food secure, mildly food insecure, moderately food insecure or severely food insecure.

Household wealth index was computed for urban and rural residents separately, using the principal component analysis (PCA). Urban wealth status was calculated by considering properties like selected household assets, whereas only tropical livestock unit was used to determine rural income residents. In the PCA, an Eigen value of >1, Kaiser-Mayer-Olkin (KMO) distribution and, in the final model, common factor scores were summed and ranked into lowest, middle, second, fourth and highest.

## Data quality control measures

The quality of data was ensured by pretesting the data collection tools on 5% of the participants in Debark district which has similar sociodemographic characteristics with Dabat district. Data were collected by 26 diploma graduate professionals and 5 individuals supervised the process. Supervision was also carried out at the spot by the Research Center's leaders. Data collectors and supervisors had a long experience working for the Dabat DHDSS site. Training was given to both data collectors and supervisors on the questionnaire, content interviewing technique, purpose of the study and on how to maintain confidentiality of information. The collected data were checked for completeness and clarity by assigned supervisors and the principal investigator.

## Data analysis procedures

Data were entered into the HRS 2 software and exported to STATA V.14 for further analysis. The overall prevalence of NCDs was defined as the presence of any of the studied NCDs during interview among the study population. The results were presented in the form of tables, figures and summary statistics. Logistic regressions were used to assess the association of various factors and the outcome variable. The bivariable analysis with a crude OR of 95% CI was used to see the crude effect of independent variables on NCDs. All independent variables that had a p value

of 0.20 in the bivariable analysis were fitted to the multivariable logistic regression analysis to control for possible confounding effects of variables. Variables which were found to be significant at a p value 0.05 level and 95% CI were considered to be the determinant factors of NCDs.

## Ethical considerations

Ethical clearance was obtained from the Institutional Review Board (IRB) of the University of Gondar before starting the actual data collection. Participants in the selected kebeles were recruited voluntarily after obtaining full information about the research and signed a written consent agreement. They were informed of their rights to withdraw from the study at any stage. Moreover, the confidentiality of information was guaranteed by using code numbers rather than personal identifiers and by keeping the data securely locked up. Individuals identified as cases of NCDs were referred to the nearby clinic for further treatment and follow-up.

## RESULTS

### Participants

A total of 67397 people were enrolled in this study. Women comprised about 50.77%. The majority of the study population (75.10%) lived in rural areas. Among the study subjects, 42.96% (28 952) were <15 years of age. Christianity is the dominant religion, accounting for 96.36% of the participants. The sociodemographic characteristics of the study population are depicted in table 1.

### Descriptive data

About 8.87% of the participants never drank alcohol in their life, while 88.2% and 2.88% were occasional and frequent consumers, respectively. About 99.5% of the study participants never smoked cigarettes in their lifetime, and 44.7% performed vigorous work-related physical activities on daily bases (figures 1 and 2).

### Outcome data

Participants who had NCDs were 1160 (1.7%) (95% CI 1.62 to 1.82), and according to the proportionate NCDs, morbidity ratio (morbidity fraction) was 404 (32.2%) heart disease, 401 (31.9%) HTN 347 (27.7%) asthma, 62 (4.9%) DM and 40 (3.2%) cancer (table 2). The mean (±SD) age of subjects with NCDs was 49.32 (±17.9). Of these, 746 (64.6%) were women with 48.96% in the food insecure group. The prevalence of all types NCDs was slightly higher among women in all age groups. Out of the 16035 households, 1146 (6.98%) had at least one NCD prevalence per household in the HDSS, and 40 (3.6%) of these had two NCDs cases per household.

### Factors

After adjusting for a number of important covariates, the multivariable logistic regression analysis showed that increasing age, alcohol consumption, high income, living in urban areas, farming, work-related physical activities and severe food insecurity were associated with the

| Table 1 Sociodemographic characteristics of the population of Dabat HDSS Research Center, northwest Ethiopia, 2016 | | |
|---|---|---|
| **Variable** | **Frequency** | **%** |
| Sex | | |
| Man | 33 181 | 49.23 |
| Woman | 34 214 | 50.77 |
| Age (years) | | |
| <15 | 28 952 | 42.96 |
| 15–24 | 13 289 | 19.72 |
| 25–34 | 8170 | 12.12 |
| 35–44 | 6322 | 9.38 |
| 45–54 | 4720 | 7.00 |
| 55–64 | 3020 | 4.48 |
| =65 | 2918 | 4.33 |
| Residence | | |
| Rural | 50 769 | 75.10 |
| Urban | 16 835 | 16.84 |
| Marital status | | |
| <10 years old | 20 089 | 29.81 |
| Married | 21 814 | 32.37 |
| Single | 19 746 | 29.30 |
| Divorced | 2390 | 3.55 |
| Widowed | 2369 | 3.52 |
| Separated | 917 | 1.36 |
| Cohabiting | 68 | 0.10 |
| Religion | | |
| Orthodox | 64 940 | 96.36 |
| Muslim | 2444 | 3.63 |
| Others (catholic, protestant) | 11 | 0.01 |
| Ethnicity | | |
| Amhara | 67 294 | 99.85 |
| Tigre | 84 | 0.12 |
| Others (Oromo, Agaw) | 17 | 0.02 |
| Education | | |
| Not on education (<7 years) | 13 672 | 20.29 |
| Unable to read and write | 21 149 | 31.38 |
| Read and write | 5541 | 8.22 |
| Grade 1–4 | 10 960 | 16.26 |
| Grade 5–6 | 4560 | 6.77 |
| Grade 7–8 | 3590 | 5.33 |
| Grade 9–10 | 4375 | 6.49 |
| Grade 11–12 | 1957 | 2.90 |
| Grade 12 and above | 1591 | 2.36 |
| Occupation | | |
| No occupation (<10 years) | 20 436 | 38.45 |
| Students | 13 955 | 26.26 |
| Farmers | 12 647 | 23.80 |
| Employed permanent | 1951 | 3.67 |
| Private job | 1167 | 2.20 |
| Job seeker | 1056 | 1.99 |
| Merchant | 656 | 1.23 |
| Housemaid | 623 | 1.17 |
| Employed contract | 328 | 0.62 |
| Retired | 296 | 0.56 |
| Others(housewife, shepherd, disabled) | 33 | 0.05 |
| | | Continued |

| Table 1 Continued | | |
|---|---|---|
| **Variable** | **Frequency** | **%** |
| Location | | |
| Low land | 22 380 | 66.9 |
| High land | 45 224 | 33.1 |
| Relation to the HH head | | |
| HH head | 16 082 | 23.86 |
| Housewife | 10 542 | 15.64 |
| Son/daughter | 34 702 | 51.49 |
| Sister/brother | 538 | 0.80 |
| Mother/father | 310 | 0.46 |
| Grandson/granddaughter | 3095 | 4.59 |
| Other relative | 1196 | 1.77 |
| Other non-relative | 930 | 1.38 |
| Family size | | |
| 1–4 | 24 512 | 36.35 |
| 5–9 | 41 667 | 61.79 |
| 10–15 | 1250 | 1.85 |
| Wealth status | | |
| Poorest quintile | 9475 | 14.58 |
| Second quintile | 11 344 | 17.45 |
| Third quintile | 13 031 | 20.05 |
| Fourth quintile | 14 593 | 22.45 |
| Richest quintile | 16 577 | 25.47 |

HDSS, Health and Demographic Surveillance System.

odds of NCDs. Advanced age (≥65 year) (adjusted OR (AOR)=19.6; 95% CI 5.83 to 65.7), living in urban areas (AOR=2.20; 95% CI 1.83 to 2.65), severe household food insecurity (AOR=1.71; 95% CI 1.37 to 2.12) and high income (the odds NCD was 28% high among the high income group compared with the poor income group) (AOR=1.28; 95% CI 1.02 to 1.59) were significantly associated with reported history of NCDs, while low alcohol consumption (AOR=0.36; 95% CI 0.31 to 0.42), moderate alcohol consumption (AOR=0.33; 95% CI 0.22 to 0.48), farming occupation (AOR=0.72; 95% CI 0.57 to 0.91) and work-related physical activities (AOR=0.66; 95% CI 0.50 to 0.88) were inversely associated with NCDs (table 3).

## DISCUSSION
### Key result
The number of NCD cases seen at the Dabat HDSS site was larger among the elderly; the proportions were considerably high among urban dwellers. A similar to another study done in rural and urban Ethiopia, older age, physical inactivity, living in urban areas, farming, alcohol consumption, severe food insecurity and low income were the strongest predictors of NCDs.[9] Similar review study done in Sub-Saharan countries showed that NCDs are now the second most common causes of death after infectious diseases.[10] Our findings show that the occurrence of NCDs is significantly high among severely food insecure groups. This study is supported by a finding from Bangladesh which showed that persons with shorter stature are more likely to develop NCDs like diabetes and

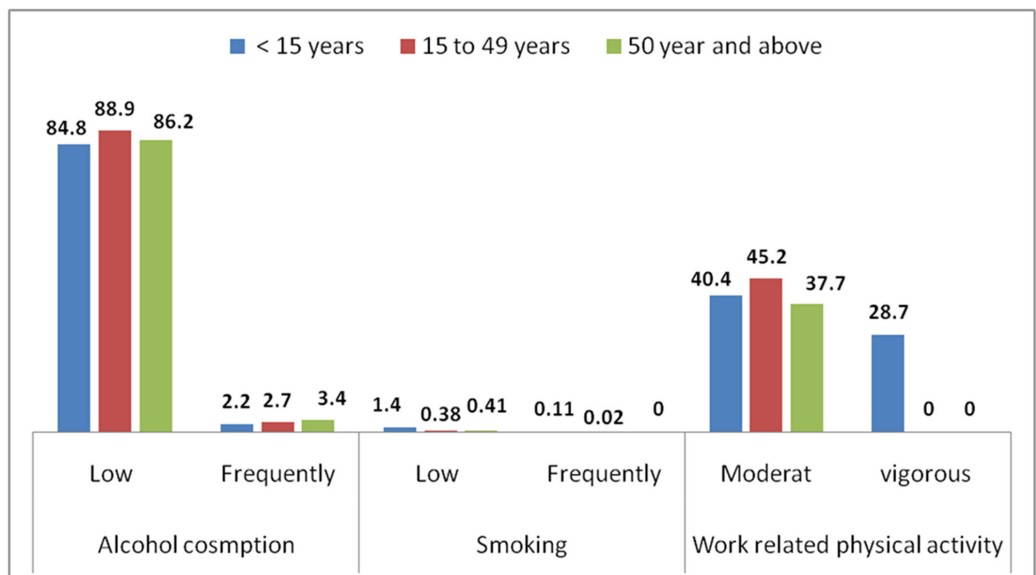

**Figure 1** Reported alcohol intake, smoking and physical activity by age group of Dabat HDSS Research Center population, northwest Ethiopia, 2016. HDSS, Health and Demographic Surveillance System.

hypertension.[11] The WHO disease classification report shows that the disease phenotype encountered in Africa is very different from the West and most closely resembles malnutrition-related. Some researchers also indicated that early exposure to cereal products or solid foods might increase immune response, which could trigger the destruction of β-cells.[12]

Another finding shows that people who were small at birth and grew least during the first year of life had a chance of developing NCDs in later life.[13] This disease burden can affect the normal growth of the people, thereby developing early onset of non-communicable diseases in early life and childhood. This exposure is likely to be responsible for the observed high rate of prevalence in this low economic status group.[14] Likewise, in our study

a high level of NCDs were noted among individuals with high income.

This can be plausibly explained by the fact that people with better economic status will be more likely to favour mechanisation which in turn leads to transition from physically active to less physically active sedentary lifestyle, keeping people indoors. Moreover, people with better economic status will be more likely to follow unhealthy eating habits as they can afford diets richer in unsaturated fat, thereby being more susceptible to obesity and development of NCDs.[15 16] These data suggest that the burden of NCDs is increasing, perhaps as a result of demographic and epidemiological changes. Even though the country is undergoing a delayed transition, the burden of both malnutrition-related and non-communicable diseases

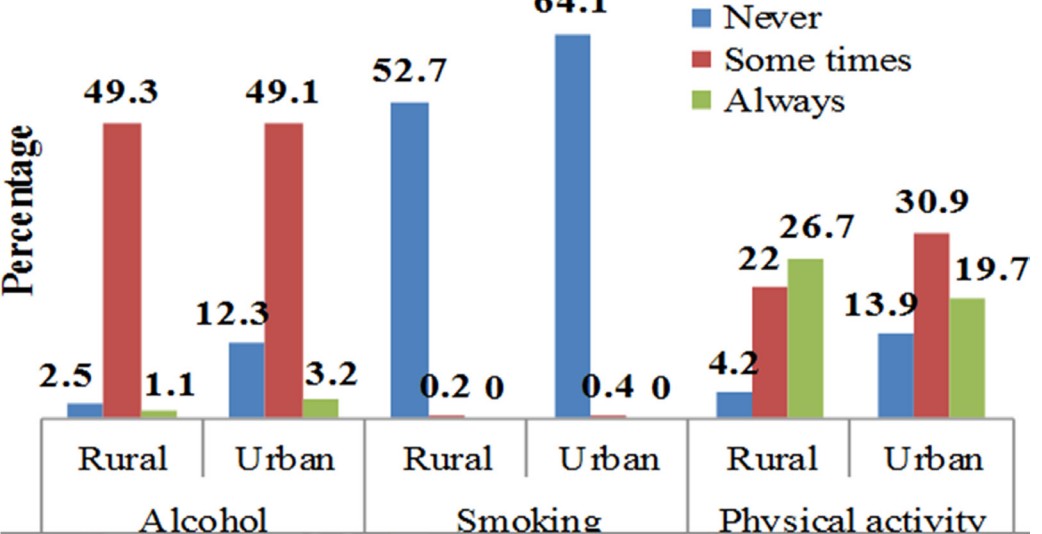

**Figure 2** Frequency smoking, physical activity and alcohol consumption of Dabat HDSS Research Center population, northwest Ethiopia, 2016. HDSS, Health and Demographic Surveillance System.

**Table 2** Proportion of NCDs by age and sex among Dabat HDSS site population, northwest Ethiopia, 2016

| Variable | Gender | Reported cases age (year), n(%) | | | | | | |
|---|---|---|---|---|---|---|---|---|
| | | 1–14 | 15–24 | 25–34 | 35–44 | 45–54 | 55–64 | ≥65 |
| DM | Woman | 1 (0.01) | 5 (0.08) | 2 (0.05) | 3 (0.09) | 3 (0.12) | 7 (0.45) | 5 (0.38) |
| | Man | 4 (0.03) | 7 (0.10) | 7 (0.18) | 5 (0.17) | 4 (0.19) | 4 (0.28) | 5 (0.31) |
| HTN | Woman | 1 (0.010) | 6 (0.09) | 22 (0.50) | 30 (0.91) | 49 (1.90) | 67 (4.26) | 86 (6.56) |
| | Man | 2 (0.01) | 4 (0.06) | 3 (0.08) | 15 (0.50) | 24 (1.12) | 29 (2.0) | 60 (3.73) |
| Heart disease | Woman | 10 (0.07) | 31 (0.47) | 54 (1.24) | 50 (1.52) | 66 (2.56) | 43 (2.74) | 47 (3.59) |
| | Man | 4 (0.030) | 12 (0.18) | 14 (0.37) | 14 (0.46) | 19 (0.89) | 12 (0.83) | 27 (1.68) |
| Asthma | Woman | 2 (0.01) | 7 (0.11) | 32 (0.73) | 41 (1.24) | 44 (1.71) | 36 (2.29) | 31 (2.37) |
| | Man | 4 (0.03) | 6 (0.09) | 21 (0.55) | 33 (1.09) | 32 (1.50) | 29 (2.0) | 28 (1.74) |
| Mental health | Woman | 12 (0.08) | 26 (0.39) | 27 (0.62) | 18 (0.55) | 19 (0.74) | 11 (0.70) | 9 (0.69) |
| | Man | 11 (0.08) | 21 (0.31) | 16 (0.42) | 14 (0.46) | 16 (0.75) | 3 (0.21) | 5 (0.31) |
| Cancer | Woman | 2 (0.01) | 0 (0.00) | 2 (0.05) | 6 (0.18) | 10 (0.39) | 0 (0.00) | 6 (0.46) |
| | Man | 1 (0.01) | 2 (0.03) | 1 (0.03) | 1 (0.03) | 1 (0.05) | 5 (0.35) | 3 (0.19) |

DM, diabetes mellitus; HDSS, Health and Demographic Surveillance System; HTN, hypertension; NCDs, non-communicable chronic diseases.

remains high.[17 18] This double burden has a negative implication on major economic transitions.[19]

The effect of ageing on NCD prevalence is already evident in Africa.[20] Ageing causes a progressive decline in the strength of musculature, which causes muscle atrophy and the risk of developing CVD. In our finding, the prevalence of NCDs increases with advancing age. This finding is consistent with other reports from India,[15] as NCD epidemic matures, the onset will shift to younger age groups. At ageing, many more people are exposed to the risk factors for long periods until the complications develop and they experience the clinical syndromes of NCDs.[17] In Ethiopia, lack of awareness, distance to facilities and unaffordability of health services delay diagnoses for the elderly.[21]

Physical activity is more common in rural than urban regions of Africa because rural dwellers often rely on intense agricultural activities as their main occupation. In line with this, our finding shows that the prevalence of NCDs is low among farmers and physically active individuals which is consistent with the meta-analysis done on the risk factors for chronic non-communicable diseases in China, Ghana, Mexico, India, Russia and South Africa.[22] Sedentary lifestyle and lack of physical activities accelerate the occurrence of NCDs; this condition is a growing challenge to sub-Saharan countries.[23 24] Walking time and space are drastically reduced in an urban community as compared with a rural community. This reduction in physical activity associated with city life, partly explains the excessive prevalence of NCDs in urban areas. Studies show that lack of physical activity appears to be a significant risk factor for CVD in sub-Saharan Africa.[15 17] Reductions in physical activity and more sedentary modes of living parallel nutrition transition, producing an overall positive energy balance result in a paradigm of lifestyle transition.[25]

In developing countries, rapid and unplanned urbanisation and modernization resulting in socioeconomic and behavioural changes in people directly affects the health of the urban population.[17 26] The main difference in physical activity between the two types of community, however, is the use of walking in rural areas as means of transportation. A diet high in total fat, cholesterol, sugar and other refined carbohydrates, and low in polyunsaturated fatty acids and fibre, often accompanied by an increasingly sedentary life, characterise life in the urban community.[26] In other presumptions people exposed to periodic famines, which frequently occur in Africa, would through natural selection increase the frequency of certain genetic trait(s); these genes, also called 'thrifty genes', will be expressed as individuals get exposed to undernutrition and starvation. This adaptation will enable individuals be able to cope with the shortage of energy by suppressing body metabolism and limiting it only to sustaining the highly vital metabolisms of the body. However, when such individuals with activated thrifty genes from undernutrition experience improvement in their nutritional intake, the already suppressed level of body metabolism may not be able to improve to metabolise the otherwise normal nutritional intake. This inability to sufficiently metabolise nutrients will lead to fat accumulation, dyslipidaemia and diabetes despite the individual's relatively normal nutritional intake. Therefore, a continued discussion of proven cost-effective interventions of NCDs will be useful for improving public health policy.[25]

### Strengths of the study

This is the first community-based study examining the magnitude and associated factors of NCDs in the wider population, HDSS site, in Ethiopia.

The findings of this study have provided important information that can be used to improve screening and

**Table 3** Factors associated with NCDs of the population of Dabat HDSS Research Center, northwest Ethiopia, 2016

| Variable | NCDs case, n(%) | COR (95% CI) | AOR (95% CI) |
|---|---|---|---|
| **Sex** | | | |
| Woman | 746 (2.18) | 1.00 | 1.00 |
| Man | 409 (1.23) | 0.83 (0.74 to 0.93) | 0.85 (0.69 to 1.03) |
| **Age (year)** | | | |
| 0–14 | 30 (0.10) | 1.00 | 1.00 |
| 15–24 | 78 (0.59) | 2.76 (2.20 to 3.45) | 1.55 (0.47 to 5.10) |
| 25–34 | 153 (1.87) | 4.17 (3.32 to 5.23) | 3.64 (1.09 to 12.13) * |
| 35–44 | 188 (2.97) | 4.73 (3.73 to 5.99) | 6.33 (1.89 to 21.2)** |
| 45–54 | 233 (4.94) | 7.23 (5.74 to 9.12) | 11.2 (3.36 to 37.5) ** |
| 55–64 | 209 (6.92) | 10.1 (7.92 to 12.79) | 16.1 (4.79 to 53.8) ** |
| >65 | 264 (9.05) | 26.27 (21.2 to 32.2) | 19.6 (5.83 to 65.7) ** |
| **Alcohol consumption** | | | |
| Never | 340 (1.03) | 1.00 | 1.00 |
| Occasional | 782 (2.35) | 0.50 (0.42 to 0.59) | 0.36 (0.31 to 0.42) ** |
| 2–3 times a week | 33 (3.04) | 0.47 (0.31 to 0.71) | 0.33 (0.22 to 0.48) ** |
| **Smoking status** | | | |
| No | 1110 (2.96) | 1.00 | 1.00 |
| Yes | 6 (3.51) | 1.73 (0.85 to 3.52) | 0.80 (0.34 to 1.91) |
| **Wealth status** | | | |
| Poor income | 170 (1.80) | 1.00 | 1.00 |
| Low income | 172 (1.52) | 0.67 (0.57 to 0.79) | 0.90 (0.71 to 1.14) |
| Mid-level income | 175 (1.34) | 0.45 (0.38 to 0.55) | 0.91 (0.72 to 1.16) |
| Better income | 279 (1.91) | 0.48 (0.41 to 0.57) | 1.19 (0.96 to 1.49) |
| High income | 336 (2.03) | 0.35 (0.29 to 0.42) | 1.28 (1.02 to 1.59) * |
| **Residence** | | | |
| Rural | 538 (1.06) | 1.00 | 1.00 |
| Urban | 617 (3.68) | 1.06 (0.93 to 1.20) | 2.20 (1.83 to 2.65) ** |
| **Education status** | | | |
| Illiterate | 10 (0.07) | 1.00 | 1.00 |
| Can read and write | 646 (3.05) | 12.54 (9.25 to 16.99) | 0.85 (0.20 to 3.52) |
| Primary school | 140 (2.53) | 6.27 (4.41 to 8.91) | 0.87 (0.21 to 3.61) |
| High school | 55 (0.50) | 2.48 (1.72 to 3.56) | 1.29 (0.31 to 5.39) |
| Diploma and above | 304 (1.89) | 3.15 (2.25 to 4.40) | 1.43 (0.35 to 5.89) |
| **Occupation** | | | |
| Under age | 556 (1.60) | 1.00 | |
| Student | 60 (0.43) | 1.55 (1.19 to 1.99) | 0.84 (0.53 to 1.34) |
| Farmer | 223 (1.76) | 3.40 (3.20 to 4.97) | 0.72 (0.57 to 0.91)* |
| Paid job | 257 (5.44) | 3.11 (2.33 to 4.14) | 1.13 (0.91 to 1.40) |
| Unemployed | 14 (1.33) | 2.92 (1.74 to 4.87) | 0.66 (0.36 to 1.20) |
| Others | 45 (13.72) | 35.7 (25.38 to 50.43) | 1.34 (0.92 to 1.94) |
| **Doing work-related physical activity** | | | |
| Under age | 287 (0.84) | 1.00 | 1.00 |
| Sometimes | 484 (2.96) | 0.24 (0.21 to 0.28) | 0.72 (0.59 to 0.86) ** |
| Most of the time | 384 (2.28) | 0.17 (0.14 to 0.19) | 0.64 (0.52 to 0.79) ** |

Continued

**Table 3**  Continued

| Variable | NCDs case, n(%) | COR (95% CI) | AOR (95% CI) |
|---|---|---|---|
| **Food security access** | | | |
| Food secure | 565 (1.77) | 1.00 | 1.00 |
| Mildly food insecure | 110 (1.51) | 1.19 (0.96 to 1.46) | 1.08 (0.86 to 1.34) |
| Moderately food insecure | 296 (1.56) | 1.71 (1.49 to 1.96) | 1.26 (1.07 to 1.48) * |
| Severely food insecure | 136 (2.37) | 2.70 (2.28 to 3.20) | 1.71 (1.37 to 2.12) ** |
| **Location of place** | | | |
| High land | 924 (2.05) | 1.00 | 1.00 |
| Low land | 231 (1.03) | 0.96 (0.85 to 1.08) | 1.01 (0.84 to 1.22) |
| **Marital status** | | | |
| Under age | 13 (0.06) | 1.00 | 1.00 |
| Married | 655 (3.00) | 4.5 (4.29 to 6.93) | 0.86 (0.20 to 3.61) |
| Single | 82 (0.42) | 4.07 (3.17 to 5.21) | 0.37 (0.09 to 1.57) |
| Divorced | 142 (5.94) | 16.1 (12.2 to 21.3) | 0.91 (0.21 to 3.88) |
| Widowed | 234 (9.88) | 23.9 (18.4 to 31.2) | 1.07 (0.25 to 4.55) |
| Separated | 29 (2.94) | 7.68 (4.94 to 11.96) | 0.69 (0.15 to 3.06) |
| **Reported history of injury** | | | |
| No | 1139 (1.70) | 1.00 | 1.00 |
| Yes | 16 (3.92) | 1.90 (1.44 to 3.82) | 2.05 (0.15 to 3.34) |

*p Value <0.05; **p value <0.001.
Other—house wife and retired.
AOR, adjusted OR; COR, crude OR ; HDSS, Health and Demographic Surveillance System; NCDs, non-communicable chronic diseases.

care through medical evaluation to detect and manage NCDs.

## Limitations

The limitation of this study is that it has used self-reported NCDs that may introduce recall and social desirability biases, and the morbidities reported might not be accurate diagnosis of the disease condition, because diseases might be miss-classified. Moreover, as this re-census is a general survey of the population, it may not provide in-depth information about specific diseases which have occurred in the community. Another limitation of this study is that it used only a cross-sectional design which might not show temporal relationships; thus, the observed associations might not necessarily be causal. Therefore, a longitudinal research is needed to assess the relationships among the variables over time.

## CONCLUSION

The number of NCD cases seen at the Dabat HDSS site is higher among the elderly, and the proportions were considerably higher among the urban population. Effective and sustainable strategy for the prevention of chronic diseases throughout, implementing new intervention strategies by providing playgrounds to the urban community, improving the livelihood of the rural community and increasing access to healthy foods are highly essential.

**Author affiliations**
[1]Department of Human Nutrition, Institute of Public Health, College of Medicine and Health Sciences, University of Gondar, Gondar, Ethiopia
[2]Department of Health Service Management and Health Economics, Institute of Public Health College of Medicine and Health Sciences, University of Gondar, Gondar, Ethiopia
[3]Department of Epidemiology and Biostatistics, Institute of Public Health College of Medicine and Health Sciences,University of Gondar, Gondar, Ethiopia
[4]Department of Environmental and Occupational Health and Safety, Institute of Public Health College of Medicine and Health Sciences, University of Gondar, Gondar, Ethiopia
[5]Dabat Research Centre Health and Demographic Surveillance System, Institute of Public Health College of Medicine and Health Sciences,University of Gondar, Gondar, Ethiopia
[6]Department of Physiotherapy, School of Medicine, College of Medicine and Health Sciences, University of Gondar, Gondar, Ethiopia
[7]Department of Reproductive Health, Institute of Public Health, College of Medicine and Health Sciences, University of Gondar, Gondar, Ethiopia

**Acknowledgements**  The authors would like to acknowledge the University of Gondar and EPH CDC for the financial support. They extend our thanks to the Dabat District Health Office for their permission to conduct the study. They want to express our gratitude to all staff of the Dabat DHS and the study participants.

**Contributors**  SMA, GA, AG, YK, TA, MW, TA, MB and KA: designed the study. SMA, AS, MB and AT: performed the analysis and interpretation of data and drafted the paper. SM, AS, MB, AT SM, GA, AG, YK, TA, MW, TA, MB and KA: revised the drafts of the paper. All authors: prepared the draft manuscript, read and approved the final manuscript.

**Competing interests**  None declared.

**Patient consent**  Obtained.

**Ethics approval**  University of Gondar IRB.

**Provenance and peer review**  Not commissioned; externally peer reviewed.

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
