## [Reviewer comments · BMJ Open]

ARTICLE DETAILS

TITLE (PROVISIONAL)	The Prevalence of Non-communicable Diseases in Northwest Ethiopia: Survey of Dabat Health and Demographic Surveillance System
AUTHORS	Abebe, Solomon; Andarge, Gashaw; Shimeka, Alemayehu; Alemu, Kassahun; Kebede, Yigzaw; Wubshet, Mamo; Woldemariam, Amare; Gebeyehu, Abebaw; Bayisa, Mulugata; Yitayal, Mezgebu; Awoke, Tadesse; Azmeraw,, Temesgen; Birku, Melkamu

VERSION 1 - REVIEW

REVIEWER	David Katende MRC/UVRI UGANDA RESEARCH UNIT, UGANDA
REVIEW RETURNED	20-Feb-2017

GENERAL COMMENTS	Minor Essential Revisions 1. Paper is interesting to read but I think the title is a bit too lengthy. Suggestion would be "The high burden of chronic Non-communicable Diseases in northwest Ethiopia: A population based survey"2. Paragraph numbering is not necessary unless a requirement of the journal.3. Results: I was not very clear on how high income was inversely associated with NCDs yet the AOR 1.28 (95%CI 1.02-1.59) – in the crude analysis it seems so COR 0.35 (95%CI 0.29-0.42) perhaps the proxy measure of income used was quite fluid and shall need to be discussed in greater detail – This is also in contrast to what is indeed stated in the abstract which states is as a direct association. While the alcohol consumption measure collected here appears protective it would be regarded as mild to moderate drinking overall; was any information collected on quantity (pints or units of alcohol) and problem/binge drinking and if so, how was it associated with NCDs prevalence.4. Minor typosa. Maintain consistency in decimal places e.g. Page 8 line 5 has 48.96% yet previously 1 decimal place has been used.b. Page 8 line 35 – the word "Large" misspelt as "largee."c. Page 8 line 41 – "high" income rather than "low" income was a strong predictor of NCDs, I thinkd. Page 9 line 19 – the word "better" misspelt as "butter"5. Referencing might need to be checked - e.g. reference 12 is not
---

	the right for the discussion on early exposure to cereal products and solid foods leading to an immune response that could trigger destruction of β-cells Level of interest 6. I think this topic is of great importance regionally and globally in light of the epidemiological transition facing much of sub-Saharan Africa. Dabat being a research experienced and routinely surveyed community I would worry about the generalisability of these findings to the general Ethiopian population. Routine surveillance would also mean that this community could also have an inherent social desirability bias in light of the many serial surveys done there and being that the most of the NCDs were noted as “reported” rather than “diagnosed” one would wonder at the accuracy of the case definitions of each the NCDs reported. What efforts were made to minimise misclassification due to case definitions of similar or overlapping conditions e.g. hypertension and cardiovascular disease? Quality of written English 7. Acceptable with correction of occasional mis-phrasings and typos
--	--

REVIEWER	Sheila Barrett Northern Illinois University USA
REVIEW RETURNED	07-Apr-2017

GENERAL COMMENTS	General Comments: Study is timely because epidemiological data are needed on this population. All findings seem to be similar to other countries both in developed and developing nations which show the gravity of NCDs and the need for interventions. Design, methods and analyses seem appropriate. Twelve authors seem like too many! See specifics below; Specifics  1. Suggest a change of title, this one is long; possibly The Prevalence of Non-communicable Diseases in Northwest Ethiopia: Survey of Debat Health and Demographic Surveillance System Abstract  1. Under objectives, remove “thereof” and write out NCDs and DHSS 2. The authors used DHSS and HDSS are they different? Stick to one abbreviation 3. Under result- start off with “One thousand....
--

4. Under conclusion- remove “therefore” in the second sentence and start with Promotion of ...
5. Change the last sentence in the conclusion after dwellers to read as “older adults and people of higher socioeconomic statuses.”

Strengths and Limitations

1. Remove “thereof” in the first bullet point
2. For the third bullet point- replace with “ Assessment of the prevalence of NCDs and associated factors”
3. Fourth bullet point- start off with “Provide important ...
4. Fifth bullet point- start with “One limitation...”

Introduction

1. Page 3- new paragraph needed at “In Ethiopia...”
2. Page 4- line 53- decision-making needs to be hyphenated and remove coma after sources on line 50
3. Page 6- line 21- write out meaning of KMO
4. Page 6- line 37- add “and how to maintain confidentiality and line 57- to control “for” possible confounding ...
5. Page 7- line 7- I believe the authors meant **Institutional** Review Board?

Results

1. Page 7- the third sentence about the participants is incomplete
2. Smoking and presence of DM seem not to be a problem in this population, how do the authors account for this in their discussion?

Discussion

1. Page 8- line 35- correct spelling of “large”
2. Line 37- start off with – “Similar to another study (9) done in rural and urban Ethiopia, older age, physical activity...predictors of NCDs. Remove the sentence about another study
3. Page 9- Lines 16-18- build the argument on higher income, The sentence about findings is consistent is hanging, remove that sentence and expand on how income is related to NCDs
4. Line 19- correct butter to better economic status
5. Lines 44-48- corrected to read “ In Ethiopia, there are lack of awareness, accessibility, distance to facilities, and the affordability of health services which will delay diagnoses for the elderly”
6. Line 57- correct spelling of meta-analysis
7. Page 10- lines 18-20-after positive energy balance, change to “resulting in a paradigm...”
8. New paragraph for “in developing countries...”
9. Need to transition from physical activity to diet so start off with “ In addition to low physical activity, a diet high in ...
10. Expand on starvation, thrifty genes and development of NCDs

	11. Start off with the strengths of the study Conclusion  1. Is NCD higher in urban population regardless of age? 2. Page 11- line 9 should read “ higher” among the elderly Acknowledgment  1. Correct to read as “We extend our thanks to Debat... “We want to express our gratitude to ...all the staff... Tables  1. Table one – location is listed twice 2. What does “not on education” represent? 3. Adjust table columns so words do not go to the next line also some words (relative, shepherd, head) are far apart 4. Adjust table columns for table 2 or landscape it. 5. Figures 1 and 2 need formatting to be more appealing
--	--

VERSION 1 – AUTHOR RESPONSE

Reviewer comments and suggestions

Reviewers comment Response to comments

Suggest a change of title, this one is long; possibly The Prevalence of Non-communicable Diseases in Northwest Ethiopia: Survey of Debat Health and Demographic Surveillance System

Answer: Thank you dear, we have accepted the comments; accordingly the title is modified based on the suggestion.

Abstract

1. Under objectives, remove “thereof” and write out NCDs and DHSS
2. The authors used DHSS and HDSS are they different? Stick to one abbreviation

Answer: We change the word, so we make a bit modification on the sentence.

Regarding the use of abbreviation, comments are well taken

The appropriate name of the study setting is Dabat Health and Demographic Surveillance System site. Therefore, we corrected it in the entire document as HDSS. Also, HDSS is spelled out at first use, objective of the abstract section.

3. Under result- start off with “One thousand....
4. Under conclusion- remove “therefore” in the second sentence and start with Promotion of ...
5. Change the last sentence in the conclusion after dwellers to read as “older adults and people of higher socioeconomic statuses.” Answer: Thank you!, corrections are made accordingly.

Strengths and Limitations

1. Remove “thereof” in the first bullet point
2. For the third bullet point- replace with “ Assessment of the prevalence of NCDs and associated factors”
3. Fourth bullet point- start off with “Provide important ...
4. Fifth bullet point- start with “One limitation... Answer: The comments are well accepted and editorial correction are done.

Introduction

1. Page 3- new paragraph needed at “In Ethiopia...
2. Page 4- line 53- decision-making needs to be hyphenated and remove coma after sources on line 50
3. Page 6- line 21- write out meaning of KMO

Answer: Now, we spelled-out the abbreviation KMO in the data analysis section.

Correction made in the revised version

4. Page 6- line 37- add “and how to maintain confidentiality and line 57- to control “for” possible confounding ...

5. Page 7- line 7- I believe the authors meant Institutional Review Board? Answer:

Correction made as suggested

Thank you! Now the mechanisms considered to maintain confidentiality of information is described well in the 'Ethical considerations' section.

Obviously, running a multivariate analysis is one of the mechanism to control the possible effect of confounders. Given that, we have described in the data analysis section about it.

-We have accepted the comment: The phrase 'Ethical Review Board' is replaced by 'Institutional Review Board'.

Results

1. Page 7- the third sentence about the participants is incomplete

2. Smoking and presence of DM seem not to be a problem in this population, how do the authors account for this in their discussion? Answer: Correction made as suggested

Hence, nine of the total (13) kebeles under the HDSS are rural. Given that, majority of the study participants are rural inhabitants where the prevalence of smoking is very low compared to the urban residents. This information is also supported by the Ethiopian Demographic and Health Survey Reports.

Also, the major livelihood of the rural residents of Ethiopia, including the current study area, is agriculture which is rain-fed and done traditional using oxen's and other animals. Therefore, they are engaged in hard physical activity to support their livelihood. Also, there poor transportation access and utilization in the rural community which might explain the lower risk of overweight and obesity as well as NCDs, including DM.

Discussion

1. Page 8- line 35- correct spelling of “large”

2. Line 37- start off with – “Similar to another study (9) done in rural and urban Ethiopia, older age, physical activity...predictors of NCDs. Remove the sentence about another study Answer: Thank you for your suggestion. We have amended the sentence accordingly.

3. Page 9- Lines 16-18- build the argument on higher income, The sentence about findings is • consistent is hanging, remove that sentence and expand on how income is related to NCDs
Correction made as suggested

Line 19- correct butter to better economic status

5. Lines 44-48- corrected to read “ In Ethiopia, there are lack of awareness, accessibility, distance to facilities, and the affordability of health services which will delay diagnoses for the elderly”

6. Line 57- correct spelling of meta-analysis Answer: All the editorial problems are corrected in the main document.

Page 10- lines 18-20-after positive energy balance, change to “resulting in a paradigm...”

8. New paragraph for “in developing countries...”

9. Need to transition from physical activity to diet so start off with “ In addition to low physical activity, a diet high in ...

10. Expand on starvation, thrifty genes and development of NCDs

11. Start off with the strengths of the study Answer: We corrected editorial problems. Also, the strength of the study is add in the main document, before the limitation section. Please see the revised version

Conclusion

1. Is NCD higher in urban population regardless of age?
2. Page 11- line 9 should read “ higher” among the elderly Answer: Yes with increasing age there is a gradual progress of NCD but the proportion was high among urban regardless of age the overall proportion in the urban was 3.68% and that of the rural was 1.06% the term 'high' is replaced by ' higher'.

Acknowledgment

1. Correct to read as “We extend our thanks to Debat... “We want to express our gratitude to ...all the staff... Answer: Thank you!, the editorial errors are corrected.

Tables

1. Table one – location is listed twice
2. What does “not on education” represent?
3. Adjust table columns so words do not go to the next line also some words (relative, shepherd, head) are far apart
4. Adjust table columns for table 2 or landscape it.
5. Figures 1 and 2 need formatting to be more appealing Correction made location deleted
2. Not on education to say according to the Ethiopian Education they are pre-school children they will not start education below this age level
3. Done according to the suggestions.
4. Done according to the suggestions.

Thank you !!!

VERSION 2 – REVIEW

REVIEWER	David Katende Scientist-Epidemiologist MRC/UVRI Uganda Research Unit Uganda
REVIEW RETURNED	23-May-2017

GENERAL COMMENTS	Minor Essential Revisions 1. Overall most revisions have been done but I think these below are still pertinent before publication e.g. in results there is need clarify how the association with high income has been reported.2. Minor typosa. Page 4 Line 23 – I think the word “disclose” was dropped for investigate as it is in the abstractb. Page 6 line 26 – the word “insured” should be “ensured”c. Page 7 line 17 – phrase “by keeping data locked” please clarify whether this means data would be encrypted and/or securely locked up.d. Page 8 First paragraph – it is difficult to understand whether this prevalence refers to the population prevalence or the household prevalence. I would advise that the sentence on households come last as a separate sentence/paragraph.e. Page 8 line 14&15 – strongly associated with – I think refers to positively associated however you have still included high income AOR 1.28 (95% CI 1.02-1.59) in those with an inverse association yet your abstract seems to state it quite rightly different as a positive association.3. Referencing shall need to be checked thoroughly as this reference [12] is not the right one for the discussion on early
--

	exposure to cereal products and solid foods leading to an immune response that could trigger destruction of β-cells – this was not corrected previously! – I would entire bibliography is rechecked. Level of interest Great local and global importance; authors have dealt with most of the issues appropriately e.g. the issues around generalisability and social desirability of a research-experienced cohort Quality of written English 4. Acceptable after recommended changes
--	--

REVIEWER	Sheila Barrett Northern Illinois University, USA
REVIEW RETURNED	04-May-2017

GENERAL COMMENTS	The comments made by the reviewers have been addressed and the paper has improved. best wishes in your research work.
---

VERSION 2 – AUTHOR RESPONSE

Reviewer(s)' Comments to Author:

Reviewer: 2

Reviewer Name: Sheila Barrett

Institution and Country: Northern Illinois University, USA Answer: Thank you dear, we have accepted the comments;

Reviewer: 1

Reviewer Name: David Katende

Institution and Country: Scientist-Epidemiologist,

Minor Essential Revisions

1. Overall most revisions have been done but I think these below are still pertinent before publication e.g. in results there is need clarify how the association with high income has been reported.

Comments are well taken: The odds NCD was 28% high among high income group compared with poor income

Increasing income is associated with the occurrence of NCD

2. Minor typos

a. Page 4 Line 23 – I think the word “disclose” was dropped for investigate as it is in the abstract
As suggested: Correction made in the revised version disclose dropped /change to assess

b. Page 6 line 26 – the word “insured” should be “ensured”

Thank you!, corrections are made accordingly.

c. Page 7 line 17 – phrase “by keeping data locked” please clarify whether this means data would be encrypted and/or securely locked up.

Answer: The comments are well accepted: to keep confidentiality securely locked up

d. Page 8 First paragraph – it is difficult to understand whether this prevalence refers to the population

prevalence or the household prevalence. I would advise that the sentence on households come last as a separate sentence/paragraph.

Answer: Thank you

Correction made in the revised version

e. Page 8 line 14&15 – strongly associated with – I think refers to positively associated however you have still included high income AOR 1.28 (95% CI 1.02-1.59) in those with an inverse association yet your abstract seems to state it quite rightly different as a positive association.

Answer: comments are well taken

Correction made as suggested

Thank you!

3. Referencing shall need to be checked thoroughly as this reference [12] is not the right one for the discussion on early exposure to cereal products and solid foods leading to an immune response that could trigger destruction of β -cells – this was not corrected previously! – I would entire bibliography is rechecked.

Answer: Correction made as suggested

Thank you !!!

VERSION 3 – REVIEW

REVIEWER	David Katende MRC/UVRI, Uganda
REVIEW RETURNED	01-Jun-2017

GENERAL COMMENTS	All my remaining concerns have been addressed. Minor typos Page 3 line 11 - In Abstract factors - ..associated with "it.Our"... should be split to read "associated with them. Our findings..". Page 5 line 48 " A structured..." - "A" is in bold it should be normal font. Good luck and all the best.
--